# To Include or Occlude: Rational Engineering of HCV Vaccines for Humoral Immunity

**DOI:** 10.3390/v13050805

**Published:** 2021-04-30

**Authors:** Felicia Schlotthauer, Joey McGregor, Heidi E Drummer

**Affiliations:** 1Viral Entry and Vaccines Group, Burnet Institute, Melbourne, VIC 3004, Australia; felicia.schlotthauer@burnet.edu.au (F.S.); joey.mcgregor@burnet.edu.au (J.M.); 2Department of Microbiology and Immunology, Peter Doherty Institute for Infection and Immunity, University of Melbourne, Melbourne, VIC 3000, Australia; 3Department of Microbiology, Monash University, Clayton, VIC 3800, Australia

**Keywords:** glycoprotein E2, vaccine development, humoral immunity

## Abstract

Direct-acting antiviral agents have proven highly effective at treating existing hepatitis C infections but despite their availability most countries will not reach the World Health Organization targets for elimination of HCV by 2030. A prophylactic vaccine remains a high priority. Whilst early vaccines focused largely on generating T cell immunity, attention is now aimed at vaccines that generate humoral immunity, either alone or in combination with T cell-based vaccines. High-resolution structures of hepatitis C viral glycoproteins and their interaction with monoclonal antibodies isolated from both cleared and chronically infected people, together with advances in vaccine technologies, provide new avenues for vaccine development.

## 1. Introduction

The development of direct-acting antiviral agents (DAA) with their ability to cure infection in >95% of those treated was heralded as the key to eliminating hepatitis C globally. Now, after more than 7 years of direct-acting antiviral therapy availability only nine countries will be able to eliminate hepatitis C by 2030 [1]. Modelling the use of a vaccine alongside DAA therapy and harm reduction strategies demonstrates that a vaccine can substantially increase the number of countries able to reach elimination by 2030 and reduce the cost of elimination strategies [2]. Most successful vaccines against viruses generate neutralizing antibodies (NAb) that block infection in the host. Theoretically, a vaccine for HCV should be possible, as ~20% of people infected with HCV are able to spontaneously clear the virus [3]. In infected people, HCV exists as eight distinct genotypes and more than 67 subtypes, and within infected people the virus continues to mutate, existing as a mixture of genetically distinct, but closely related, variants or quasispecies [4,5]. Prior infection with one genotype does not necessarily prevent re-infection even with a closely related virus suggesting that natural immunity may be short-lived or narrowly focused. A successful vaccine for HCV will need to afford protection against all eight highly divergent genotypes of HCV. Ideally, a vaccine would confer ‘sterilizing’ immunity by inducing potent NAb responses towards the HCV envelope proteins of all genotypes/subtypes/quasispecies and a multi specific cellular immune response including both CD4+ and CD8+ T cells. In such a scenario, a prophylactic vaccine would prevent infection completely. Alternatively, it may be sufficient that the protective efficacy of the vaccine is measured by its ability to prevent chronic HCV referred to as clinical protection. In this scenario, vaccinated individuals exposed to HCV may have a self-limiting infection that is cleared over time and do not develop chronic HCV. Whilst early efforts to define the correlates of immunity focused on T cells and NK cell functions, more recent work has defined the role that B cells and NAbs play in controlling HCV infection and how different vaccine candidates generate different antibody specificities.

Hepatitis C virus (HCV) is a member of the Flaviviridae family of positive sense, single stranded RNA viruses that infect human hepatocytes. Its 9.6 kb RNA genome encodes for 3 structural proteins which are essential for viral entry and 7 non-structural proteins which serve as replication factors. The two viral structural proteins, E1 and E2, function as a heterodimer and facilitate viral entry [6,7]. The entry pathway of the virion into hepatocytes is a well studied complex process that requires coordinated interactions between E1 and E2 and host cell surface receptors which include host cell surface receptor CD81 and cell entry factors including scavenger receptor B1 (SR-B1), occludin and claudin-1 [8]. Antibody responses are readily generated in natural infection to both E1 and E2, although the majority of NAb are directed towards E2. Nevertheless, a subset of rare antibody specificities recognizes complex epitopes only present on the E1-E2 heterodimer. The epitopes themselves can be defined as either simple or continuous where they can be represented by a synthetic peptide or denatured antigens, or discontinuous where the epitope is only present in the context of the tertiary or quaternary structure of the protein. Neutralizing antibodies can either be type-specific, defined as the ability to neutralize the same or closely related isolates within a subtype, or broadly neutralizing when they have the capacity to cross neutralize two or more of the 8 known genotypes.

A vaccine for the prevention of HCV has remained elusive due to multiple confounding factors. Major challenges include its high level of sequence heterogeneity that translates into antigenic variation meaning a vaccine must generate protective immune responses that target conserved regions of the virus, and the lack of a simple animal challenge model that faithfully recapitulates natural infection in which to test preclinical vaccine efficacy. Further, the viral glycoproteins that are targets of the NAb response, known to prevent infection in vitro, and associated with viral clearance in vivo, possess multiple immune evasion mechanisms that suppress the generation of robust broadly reactive neutralizing (brNAb) responses. These factors together with a preconception that DAAs will be sufficient to reach elimination resulting in a lack of investment in HCV vaccine research have stalled HCV vaccine development. However, recent progress in understanding viral glycoprotein structure and how antibodies interact with the viral glycoproteins, suggest a pathway to vaccine development using novel technologies and protein engineering.

## 2. The Viral Glycoproteins

The surface of the virion is decorated by the two viral glycoproteins E1 and E2 in the form of heterodimers. Glycoprotein E1 is a 31–35 kDa protein composed of residues 191–383 (prototype H77 numbering used throughout), contains 8 highly conserved Cys residues, and is post-translationally modified through the addition of 4–5 N-linked glycans. Heterodimerization between E1 and E2 is primarily mediated through the transmembrane domain of both glycoproteins involving G354, G358, L370, and N728 [9]. Hydrophobic regions within the ectodomain of E1 have been proposed to be contributors to membrane fusion events post-receptor binding, although structural evidence is lacking to support these observations [10,11,12]. Glycoprotein E1 in itself does not appear to be a major target of the NAb response in natural infection; instead, some NAb have been identified that require a heterodimer between E1 and E2 to form with antibody contact residues identified from within E1 [13]. The exception is monoclonal antibody HEPC112 isolated from a human that developed chronic HCV that revealed a new antigenic site AS112 within E1. Residues E215, E232, G233, V246, R249, K252, R259, R260, D263, R297, W299, G354, Y361, F378, and D382 contributing to binding all located with the E1 region. HEPC112 was able to neutralize 7/24 diverse HCV strains tested showing a moderate level of breadth but did not meet the threshold for broadly neutralizing [14]. In addition, IGH505 that requires the 313–327 region of E1 and neutralizes genotypes (G) 1a, G1b, G2a, G4a, G5a and G6a [15].

Glycoprotein E2 is 70 kDa in size and contains a receptor binding domain (RBD; residues 384–661, E2_661_), a stem region (residues 662–714) and a transmembrane domain (residues 715–746) (6). Within the E2 RBD there are 3 hypervariable regions (HVR) 1, HVR2 and variable region 3 (VR3 or igVR), corresponding to residues 384–410, 459–486 and 570–580, respectively [16] and contribute to the high sequence heterogeneity of HCV. The CD81 binding region on E2 is composed of 4 discontinuous regions that include residues identified in mutagenesis studies to substantially contribute to CD81 binding are W^420^-H, L^441^-F^443^, residues Y527, W529, G530, and D535 and Y^613^RLWHY [17,18,19,20]. Contributing to the mass of E2 are 11 conserved N-linked glycans, that play a fundamental role in protein folding, viral entry and modulate recognition of antibodies [21,22,23]. E2 contains 18 conserved cysteine residues that participate in both inter and intramolecular disulfide bonding [24].

### 2.1. Structural Insights into Glycoprotein E2

Several high-resolution structures of E2 in complex with monoclonal antibodies (MAbs) have been determined ranging from the first core domain [20,25] structures to near intact ectodomain structures [26,27,28]. The first structural studies by Kong et al. (PDB: 4MWF), used a G1a minimal core domain (residues 412–645) with the removal of residues surrounding HVR2 (residues 460–485) and glycans at N448 and N576, in complex with the antigen binding fragment (Fab) of neutralizing monoclonal antibody (nMAb) AR3C. This revealed a central immunoglobulin β sandwich, flanked between a front and a back layer, stabilized by 8 disulfide bonds. Notable features include extensive disorder at the N-terminus (residues 412–420) in regions adjacent to the truncated HVR2 and the igVR, and the N-linked glycans [20]. The 4MWF structure characterizes the epitope of AR3C as being relatively flat and encompassing residues in domain E and a portion of the CD81 binding loop on the front layer of E2. The antigenic domains of E2 are shown in Figure 1 and discussed in detail in Section 2.2.

Several key residues involved in E2-CD81 interaction were identified to map to the AR3C binding site and also in one side of the beta-sandwich. This AR3C and CD81 binding site is a critical region in immunogen design as it was found to be hydrophobic, of low sequence variability and free of N-linked glycans. The core domain structure reported by Khan et al. (PDB: 4WEB) from a G2a deglycosylated E2 (residues 456–656) in complex with non-neutralizing antibody (non-NAb) 2A12 revealed a similar overall structure to Kong et al., with a distinct disulfide bonding pattern and only three disulfide bonds present in both structures. The different structural features observed by Khan et al. may be related to one or more of the highly minimized core domains lacking the entire N-terminal region to residue 456, the different sequence genotype and/or the stabilization of the core domain with a non-NAb towards the back face of E2 or the lack of glycans. These structures were the first to reveal the location of the CD81 binding site on the surface of E2 within the front layer, the presence of a “CD81 binding loop” (residues 519–535) and the discovery that NAbs bind the front face of E2, with the back face representing the non-neutralizing face of E2. 

Recently, further structures were resolved providing a more complete picture of the structure of the E2 ectodomain; G1a E2 residues 412–645 (PDB: 6MEK) and residues 384–645 (PDB: 6MEJ), in complex with nMAb HEPC3 and non-nMAb HEPC46, G1b E2 residues 384–645 in complex with nMAb HEPC74 (6MEH) and E2 residues 384–645 in complex with nMAb HEPC3 (PDB: 6MEI). Representing the most complete structures to date, they reveal that HVR2 is a flexible loop that wraps around VR3/igVR on the back face of E2 and that the N-terminal region to 420 is highly flexible and adopts different conformations. Once again, a distinct disulfide bonding pattern was observed. However, four disulfide bonds, C429–C503, C494–C564, C508–C552, C607–C644 are conserved across structures and therefore likely to be critical for E2 folding. Whether these differences in disulfide bonding between structures are related to the use of different HCV sequences, binding by different antibodies or naturally occurring disulfide exchange is unknown. Previous studies using cell culture derived HCV reveal that free thiols were essential for infectivity suggesting that thiol exchange was important for HCV entry [29].

Structural studies on 3 G6a (HK6a) E2 core domains (PDB: 6BKB, 6BKC, 6BKD) in complex with nMAbs AR3A, AR3B and AR3D, respectively, showed that the overall core structure is similar to that of the core structures of G1a (H77c) and G2a (J6) suggesting broad conservation of the core domain. However, the connecting loop of β6 and β7 of the immunoglobulin sandwich differs between G1a and G6a, possibly a result of the loss of a glycosylation site at position 540 in G3 and G6. Additional differences were noted in the VR3/igVR region and in the back layer at residues 629–640. Further examination of this difference in beta sheet structure was performed by comparing structures of G1a and G6a E2 core domains (residues 412–645) lacking HVR2 (residues 460–485), VR3 (residues 570–596), and mutations to remove glycosylation sites at N488 and N576 in complex with nMAb U1 Fab (G6a, PDB: 6WO3), HC11 fab (G6a, PDB: 6WO4), 212.1.1 Fab+E1 fab (G1a, PDB: 6WO5) and HC1AM Fab plus E1 Fab (G1a, PDB: 6WOQ). 

### 2.2. E2 Antigenic Domains Eliciting NAb

High-resolution structures of E2 have enabled a deeper understanding of the role of the variable domains, glycans and other structural elements on the antigenic properties of E2. Combined with studies of naturally occurring nMAbs isolated from acutely and chronically infected individuals, this has improved our understanding of the design of vaccines to generate desirable antibody specificities. Multiple nomenclatures for these epitopes are in use. The four most commonly used nomenclatures are antigenic domain A–E, epitope I–III, various antigenic sites and antigenic Region 1–5 (AR 1–5) with some of these being overlapping domains (Figure 1).

**Hypervariable Region 1 (HVR1)** located at the N-terminus of the mature E2 protein (residues 384–410) represents the most variable region with the HCV genome. The complete structure of intact forms of E2 containing HVR1 remain unsolved and only partial structural data is available for residues 405–410 lacking secondary structure [27]. HVR1 is highly immunodominant in both natural infection and in vaccination studies [30]. HVR1 is not essential to virus replication, although viruses lacking HVR1 are attenuated [31]. Despite this, HVR1 has been shown to play an important role in binding virions to glycosaminoglycans on the cell surface [32] and contributes to binding of virus entry co-factor SR-B1 [33]. Antibodies against HVR1 are the only type known that can interfere with binding to SR-B1 [33] and such specificities could be important contributors to virus neutralization. Consistent with this, antibodies to HVR1 have been shown to protect chimpanzees from infection with homologous HCV [34]. However, neutralizing antibodies targeting HVR1 are largely type specific and so failed to protect chimpanzees against heterologous challenge [34]. Because antibodies to HVR1 mediate potent type-specific neutralization, this drives the evolution of immune escape variants that evade such specificities resulting in a rapid diversification of viral quasispecies within infected humans [35,36,37]. The immunodominance of a non-essential region of E2 has been postulated as an immune evasion mechanism employed by HCV to confound antiviral immunity [38].

In addition to direct immune escape, HVR1 physically shields other epitopes on the E2 surface [39,40]. This is supported by studies showing that viruses lacking HVR1 are more susceptible to neutralization by polyclonal immune serum [31]. Studies examining an immune escape variant at the adjacent domain E residue at position 415 revealed a global change in antibody neutralization and postulated that amino acid changes in domain E could alter the conformation of HVR1 and alter susceptibility to neutralization [41]. A second mechanism involves interference by HVR1-directed antibodies that prevent binding of antibodies to neighboring epitopes, in particular the adjacent domain E [42]. The HVR1 directed antibody used in this particular study, diminished binding by antibodies targeting domain E, B and D but not C [42]. However, HVR1 specific antibodies may also act to enhance the breadth of neutralization when combined with other antibody specificities [43]. The combination of an HVR1 directed nMAb with a second nMAb targeting AR3 mediated neutralization of 9/11 viruses from a diverse panel of pseudotyped HCV particles, exceeding neutralization observed with each antibody in isolation [43]. Therefore, it is not clear whether the HVR1 region is beneficial to vaccine strategies. Given antibodies to HVR1 remain the only specificity that block interactions with GAGs and SR-B1 and some specificities can act synergistically, further exploration of vaccine strategies with and without HVR1 are worth consideration.

**Domain E**, also designated epitope I or AS412, resides (residues 412–423) directly adjacent to HVR1. Domain E contains conserved CD81-binding residues (W420, H421) and one potential glycosylation site [18,19]. Domain E is the target of multiple well-studied cross-genotype NAb isolated from experimentally vaccinated rodents and from naturally infected humans. In humans, bNAbs are rarely generated in natural infection, indicating the low natural immunogenicity of this region [44,45]. 

Structural studies using peptides and near intact E2 ectodomains in complex with MAbs have revealed that domain E is highly conformationally flexible. Three main conformations have been described: a beta-hairpin conformation [27,41,46,47], an extended or open conformation [48] and a semi-open conformation [49,50]. The extensive flexibility has been proposed to contribute to the low immunogenicity of domain E by reducing its ability to engage with B cell receptors, suppressing its immunogenicity and thereby contribute to immune evasion [48]. 

Escape mutations, N415K/D and N417S, have been described in chimpanzees, as well as in liver transplant patients treated with the domain E antibody HCV1 [51,52]. Mutations at N415 directly alter the epitope recognized by the majority of antibodies towards domain E, with the exception of HC33.1. The mutation N417S, while not directly involved in the epitope recognized by domain E antibodies, leads to a shift in glycosylation from N417 to N415, which in turn confers resistance to multiple domain E antibodies, again with the exception of HC33.1 [50,52,53,54,55]. The unique features of HC33.1 which recognizes a semi-open conformation of domain E that does not involve N415 could therefore be beneficial for a vaccine [50]. Cell culture derived viruses containing N415D or N417S have increased sensitivity to neutralization by nMAbs targeting other epitopes/domains within E2 and E1/E2 as well as to HC33.1 [41]. This is likely the result of structural changes in the E1/E2 complex, possibly involving the displacement of HVR1, and could be an avenue which allows exposure of vulnerable sites of HCV and exploiting a natural occurring escape mutation [41]. Theoretically, this approach would still allow the generation of antibodies with HC33.1-like specificities, and possibly increase titres of antibodies to other neutralization domains. Alternatively, vaccine approaches that generate polyclonal immune responses targeting domain E and at least one other neutralization domain would be desirable; if escape mutations occur within domain E, alternate specificities would have increased virus neutralization capacity. 

Overall, antibodies against domain E are desirable in the context of a polyclonal immune response, as they can achieve a considerable breadth of neutralization and escape mutations increase susceptibility towards other antibodies [41]. The sub-dominance of domain E highlights the necessity for a rational approach to HCV vaccine design to convert this region into a dominant antigenic domain.

**Domain D** is defined by antibodies that bind to the region encompassing the residues 420–428, 441–443 with structural contributions from residues 613–618 and is also referred to as epitope II and AS432 [56,57] and partially overlaps with AR3. Both neutralizing and non-NAb have been isolated to this region and structural data suggest it exhibits at least two distinct conformations [58,59]. Neutralizing monoclonal antibodies targeting domain D inhibit CD81 interactions, and block virus attachment and entry [56]. This is consistent with Trp-437, Leu-438, Leu-441, and Phe-442 directly contributing to interactions with the CD81 large extracellular loop and the overall high sequence conservation in this region [17]. The approach used by Keck et al. to isolate antibodies towards domain D involved introducing mutations at Y632A to suppress binding by non-NAb or D535A which silenced a dominant neutralizing epitope within domain B [56]. These antibodies show broad and uniform neutralization against multiple genotypes and no escape mutations could be selected through passaging of virus in the presence of antibody [56]. Moreover, although these antibodies recognize conformational epitopes, they are able to bind a synthetic peptide encompassing residues 434–446 [56], illustrating the potential for a peptide vaccine approach to generate antibodies specific to this region. While non-NAb against domain D have been reported, the characteristics of NAbs (potent, uniform cross-neutralization, lack of escape mutants, recognition of a peptide) to this domain suggest it is a promising and important component of an HCV vaccine candidate. However, the need to suppress binding by non-neutralizing and domain B antibodies to isolate antibodies that target domain D could be an indication that the region is subdominant with low immunogenicity in natural infection [56]. Examination of an individual who repeatedly cleared HCV infection revealed that the earliest response was towards epitopes in the 434–446 domain D region suggesting that early induction of such specificities may play an important role in clearing virus [60]. Domain E and domain D antibodies can also act synergistically to mediate enhanced CD81 inhibition and virus neutralization suggesting that simultaneous binding by these antibody specificities is possible and desirable in a vaccine context [61].

**Antigenic Region 3 (AR3)** encompasses the front E2 surface layer and includes residues from within residues 427–443 and 529–530 [62] and thus overlaps with both domain B and domain D. Residues 529–535 are also designated epitope III and contain some of the key residues for CD81 binding (Y527, W529, G530, D535). AR3 is highly immunogenic and is a major source of bNAbs in natural infection [62]. Antibodies targeting AR3 have been isolated from a number of HCV infected individuals, both during chronic infection and after viral clearance [14,62,63,64,65]. AR3 antibodies conferred partial protection in mice from HCV challenge [62]. When AR3 antibodies are used in combination with AR4 specificities, they protect mice from challenge with infectious HCV and can clear established infection in hepatocytes [66] as well as directly contribute to clearance in HCV infected individuals [67,68]. Several studies into viral resistance found that acquisition of escape mutations in AR3 either did not occur or lead to loss of viral fitness [64,67]. Investigation of the viral escape mechanism suggested that escape mutants could develop within the 425–443 region, but residues 529–535 did not tolerate mutations that retain fitness. The breadth of neutralization of AR3 antibodies, their common occurrence in natural infection and their contribution to resolution of infection, highlights the importance of this epitope for vaccine design. However, the potential for viral escape mutations will have to be kept in mind, so a vaccine eliciting AR3 type antibodies, should elicit other antibody species as well, to prevent selection of escape mutants.

Interestingly, several human nMAbs targeting AR3 on HCV E2 are derived from the VH1-69 germline (AR3-type, HC84-type, HepC3/74-type, HC-1-type) [13,56,62,69,70]. The use of this gene in nMAbs has been described for other viruses such as HIV and influenza [71,72,73,74]. One main mode of interaction of these antibodies with the antigen is through a hydrophobic tip of the heavy chain CDRH2, which interacts with a hydrophobic pocket on the E2 surface [26,28]. Another distinct feature in some of VH1-69 antibodies is a longer CDRH3, which can help access epitopes which are obscured on the antigen surface by mechanisms including the presence of glycans. Surprisingly, these antibodies achieve their breadth through rapid maturation with little somatic mutation [63,70]. Overall, AR3 is desirable for inclusion in vaccine candidates, with the generation of VH1-69 specificities representing an additional benefit. 

**Domain B** specific antibodies are also conformation dependent and mostly neutralizing and inhibit E2 binding to CD81. Mutagenesis studies show that two conserved E2 residues at G530 and D535 are required for binding by all neutralizing domain B MAbs. In addition, some domain B antibodies require G523 or W529 [69,75,76] and residues from within the regions 424 to 431 and 436 to 443 for binding [64]. Competition assays revealed that domains C and B do not overlap [77,78]. As such, domains B and D overlap with shared residues in their epitopes at 441–443, and both antigenic domains B and D are included in AR3.

**Domain C** specific antibodies are conformation dependent, neutralizing and inhibit binding between E2 and CD81 [66]. The earliest example is CBH-7, an antibody isolated from a patient with asymptomatic G1b infection [77,78]. The activity of CBH-7 was reported to increase when pseudotyped particles bearing E1E2 heterodimers were treated at low pH [79]. Blocking experiments using AR3 and AR5 specific MAbs AR3A and AR5A reveals that their epitopes at least partially overlap with CBH-7 [13]. Analysis of a panel of alanine mutants across the E2 ectodomain revealed that residues W549 and N540 contribute to the epitope recognized by CBH-7 [80,81]. Rodent antibodies have also been isolated whose epitopes overlap with domain C including MAb44 that inhibits CD81 binding and has type-specific neutralization of G1a pseudotyped viruses [80].

**AR4 and AR5** were discovered using “exhaustive panning of a phage-display antibody repertoire” to discover rare E1E2-specific MAbs [13]. Three unique specificities were discovered that only recognize their epitopes in the context of E1E2 heterodimers and do not inhibit E2-CD81 interactions. Competition studies show that AR4 is an entirely unique domain on E1E2 while AR5 overlaps with domain C [13]. Mutagenesis in the context of E1E2 revealed that MAb AR4A requires highly conserved amino acid D698 within the membrane proximal external region of E2, while MAb AR5A requires highly conserved residue R639 for binding. MAb AR4A has extremely broad and potent neutralization activity against a diverse panel of HCV isolates across the 6 major genotypes. By contrast, neutralization activity of AR5A was variable and more restricted to G1, G4, G5 and G6 [63]. Using genetically humanized mice permissive to limited HCV infection, prophylactic administration of AR4A was able to reduce infection with G1b and G2a cell culture derived viruses [63]. A combination of AR3A, AR3B and AR4A was able to prevent infection of humanized mice and abrogated persistent infection of hepatocytes [66].

### 2.3. E2 Antigenic Domains Eliciting Non-NAb

Some antigenic domains on the E2 surface elicit non-NAb, and their epitopes are of just as much of importance in vaccine design as they can be immunodominant acting as decoys and suppress the induction of NAb. Rational vaccine design can be used to either remove or alter non-neutralizing epitopes to suppress the generation of non-NAbs and enhance induction of NAbs. Non-NAb can interfere with the actions of NAb in multiple ways. In the HIV field it has been shown that high affinity non-NAb can prevent antigen binding to B cell receptors with specificity for bNAb epitopes [82]. As mentioned above non-NAb towards HVR1 can also interfere more directly with NAb to other epitopes by steric hindrance [42]. Epitopes targeted by non-NAb are predominantly located on the back layer of E2, which has also been termed the non-neutralizing face and is in close proximity to the CD81 binding loop. 

**Domain A** is located on the back layer of E2 (residues 581–584, 627–633) and is highly immunogenic [77,78,83]. However, a study found that the introduction of alanine mutations in this region can abrogate binding of non-NAbs and in this way shift the immune response away from these non-NAb [81]. 

**AR1** consists of a collection of residues close to the CD81 binding loop and AR3 (residues 495, 519, 544, 545, 547, 548, 549, 632) with variable ability to block E2-CD81 binding [62]. AR1 residues appear to be accessible on soluble E2 protein, but were occluded on the virion surface [62]. Studies in rodents vaccinated with recombinant E2 have also identified non-NAb with epitopes on the non-neutralizing face of E2 [80].

**Other non-NAb targets**. Non-NAbs have also been identified that were in close proximity to the CD81 binding loop [80]. One characteristic that distinguished these non-NAbs from NAbs that had overlapping epitopes was the inclusion of Y527 and W529 in the epitopes of the non-NAbs [80]. It is possible that these antibodies bound to E2 in such a way that that did not result in blockade of E2 binding to CD81 and consequent virus neutralization, or the affinity of these antibodies was below that required to efficiently neutralize virus. Affinity maturation of antibodies in vitro has been used to enhance the properties of MAbs and improve neutralization suggesting that some non-NAbs lack activity because of low affinity [49,84].

Additionally, some antibody specificities can interfere with the neutralizing activity of other NAbs. A study found that immunization of healthy adults with recombinant E1E2 can induce both epitope I and epitope II specific antibodies. Removal of epitope II specific NAbs from the immune serum increased the neutralization activity by up to 4.9 fold [85]. This was also observed in chimpanzees immunized with E1E2, whereby removal of epitope II specific antibody not only enhanced neutralization activity of serum, but also revealed a cross neutralizing antibody response [86]. This suggests that while both epitope I and epitope II specific antibodies are induced, not all epitope II directed antibodies are indeed neutralizing or beneficial to neutralization activity of serum. 

### 2.4. Polyfunctional Antibodies in HCV Infection

In addition to their neutralizing capabilities, antibodies can also mediate other functions through their Fc domain. These include antibody—dependent cellular cytotoxicity (ADCC). Very limited studies are available on the role of such polyfunctional antibodies and ADCC in HCV infection. Studies of HCV-infected individuals have shown the presence of anti-E2 antibodies capable of mediating ADCC [87]. However, these types of antibodies were more prevalent in chronic infection than in acute or self-limited infection, arguing against a contribution to resolution of infection [87]. Given that the hepatitis C E1 and E2 glycoproteins are strongly retained in the endoplasmic reticulum during biosynthesis, it is not whether ADCC plays a major role in immune clearance of HCV infection. Further studies are warranted to resolve this question. 

## 3. Vaccine Research and Strategies

Multiple strategies have been employed in an effort to generate broad and potent neutralizing antibodies towards HCV. The majority of these vaccine candidates have only been tested in preclinical studies conducted in small animals and new tools now enable are far more detailed understanding of the antibody specificities that are generated, and the breadth of neutralization of the immune serum using diverse panels of envelope proteins in pseudotyped viruses and cell culture derived HCV. 

### 3.1. Recombinant Protein Vaccines

Protein vaccines are typically composed of either purified proteins obtained from a pathogen, or recombinant expressed proteins of the pathogen. Our knowledge of E2 and the epitopes recognized by NAbs highlights the need for a rational approach of recombinant protein design for vaccine antigens. Many of the epitopes of NAbs are of low immunogenicity because they are present in regions that display high levels of conformational flexibility, and/or are shielded by other regions of E2 suppressing their ability to engage B cell receptors. As knowledge of the structure of E2 improves, vaccine approaches aimed to generate recombinant E2 vaccines have become more rational and refined.

Confounding the expression and purification of full length E1 and E2 heterodimers as vaccine candidates is the presence of ER retention signals in the transmembrane domains of both glycoproteins meaning that purification must occur from within cells, or the TMDs must be replaced with alternate sequences allowing secretion but retaining heterodimerization. The benefit of rE1E2 vaccines is that they might more closely represent the native conformation of the proteins on the virion surface, including the formation of the AR4 and AR5 antigenic domains. However, this approach may lead to the presentation of immunodominant non-neutralizing epitopes and/or dominant type-specific neutralization domains that fail to confer broad protection, such as HVR1, and does not overcome issues of conformational flexibility or epitope shielding. 

The earliest study to demonstrate that recombinant E1E2 could be a vaccine candidate was Choo et al. [88] where vaccina virus encoding E1E2 was used to infect HeLa cells and E1E2 heterodimers were co-purified and used to immunize chimpanzees. Five of 7 chimpanzees were protected from homologous HCV challenge while 2 chimpanzees had a blunted infection. Subsequent studies used recombinant E1E2 (rE1E2) produced in stably transfected CHO cells containing E1E2p7 or E1E2 [89]. Proteins produced in these cells formed E1E2 heterodimers, were retained intracellularly, and were modified by the addition of high mannose N-linked glycans. The heterodimers were recognized by MAbs specific to domain B [89]. Immunization studies conducted in mice and guinea pigs showed that both E1E2 and E1E2p7 generated robust antibody responses and homologous neutralization activity, comparable to sera obtained from animals vaccinated with E2 alone. Sera from animals vaccinated with E1E2p7 demonstrated cross-neutralization against G1a, G1b, with lower activity observed against G2a. The mechanism of neutralization was attributed to the ability of the immune serum to prevent E2 binding to CD81, although no further analysis of antibody specificity was performed [90]. With these promising results, the rE1E2 vaccine was assessed in a phase I human clinical trial. The study comprised 16 adults per group and received four doses of either 4 µg, 20 µg or 100 µg rE1E2 formulated with the human adjuvant MF59. Recipients began to seroconvert after the second dose of vaccine and there were no significant differences in geometric mean antibody titres between the groups that received different doses [91]. Antibodies able to prevent E2 interacting with CD81 were generated in a proportion of subjects with the largest numbers detected two weeks post 3rd vaccination for the 4 µg dose (71%), the 20 µg dose (79%) and the 100 µg dose (75%). The specificity of the immune response was evaluated using biotinylated peptides representing continuous epitopes on E1 (residues 313–327), HVR1 (384–410), domain E (epitope I, 412–419) and domain D (epitope II, 434–446) [92]. Results showed that volunteers had made antibodies reactive to these domains with the highest proportions towards domain E > HVR1 > E1 > domain D [92]. It is important to note that the domain E peptide did not include the critical W420 residue that is a common feature shared with all neutralizing domain D antibodies isolated to date and so this may have resulted in an underestimation of the presence of such specificities. Neutralizing antibody activity was measured in three different assays and found that 10/41 samples tested neutralized chimeric VSV-HIV pseudotypes, 15/36 neutralized HCV-HIV pseudotypes and 10/36 neutralized G1a cell culture grown virus [92]. The presence of NAb was subsequently examined using a well described HCVcc assay and found NAb activity in 12/13 volunteers examined after vaccination with the three subjects with the highest level of NAb showing cross genotype neutralization; two of the subjects were able to neutralize all 7 genotypes [93]. No phase 2 studies were conducted to assess safety and efficacy. However, the data clearly show that NAbs can be generated using this approach. 

To overcome the difficulties associated with purifying intracellularly retained E1E2, an Fc tag was added at the junction between E1 and E2 allowing purification of heterodimers using protein G Sepharose, and removal of the Fc domain with a protease [94]. Purified heterodimers were efficiently recognized with antibodies targeting AR3, AR4 and AR5 as well as domain E and D. Mice immunized with purified heterodimers generated high titre antibodies, neutralized G1a efficiently and showed cross-neutralization towards G5a. The specificity of the antibody response was examined and revealed antibodies towards AR3, AR5 and domain D with only limited amounts of antibody targeting domain E and AR4. The ability to generate novel specificities targeting the highly conserved AR5 region suggests this approach may overcome limitations associated with vaccination with E2 alone.

Evidence that HVR1 is immunodominant, can shield epitopes present on the core domain of E2 and interfere with the binding of some epitope I directed antibodies led researchers to explore whether deletion of HVR1 from rE1E2 vaccines could be a superior vaccine candidate [95]. Removal of HVR1 from rE1E2 (rΔHVR1E1E2) did not alter expression or the ability of E1E2 to form heterodimers. Antibodies directed to AR3, domain D, AR4 and AR5 recognized rE1E2 and rΔHVR1E1E2 similarly, but domain E/epitope I directed antibodies showed reduced binding to rΔHVR1E1E2. Mice immunized with rE1E2 and rΔHVR1E1E2 had similar titres of antibodies, but rΔHVR1E1E2 immune serum displayed lower neutralization activity towards homologous virus consistent with a major role for HVR1 directed antibodies to mediate type-specific neutralization. However, cross-neutralization of G3a, G4a and G5a viruses was similar for both groups of immune sera suggesting that whilst removal of HVR1 may decrease type-specific NAb activity, the generation of cross NAbs is not affected, but they are also not enhanced by removal of HVR1. Similar results were observed in guinea pigs. Aside from the lack of HVR1 specific antibodies in rΔHVR1E1E2 vaccinated animals, the specificity of antibodies was similar to rE1E2 immune serum, with AR3, domain D, E, AR4 and AR5-like specificities generated [95]. 

Hypervariable region 2 and igVR/VR3 have been demonstrated to contribute to epitope shielding and represent flexible regions of the E2 proteins [80]. The removal of all three variable domains simultaneously from the E2 RBD construct (residues 384–661; Δ123) did not affect the expression, global folding of E2 or its ability to bind CD81 suggesting that the variable domains do not contribute to formation of the CD81 binding site on E2 [16]. Alhammad et al. examined the contribution of HVR1, HVR2 and igVR/VR3 to the exposure of antigenic domains on E2 by individual and combined removal from E2_661_ and observed that all three variable domains contribute to the exposure of the epitopes of some antibodies including those directed towards domain D and AR3 with 2–3-fold increases in binding when all three variable regions were deleted. Immunization studies conducted with Δ123 in guinea pigs using three vaccinations of 100 µg in Iscomatrix^®^ revealed that Δ123 vaccinated animals generated NAbs towards homologous HCV, but titres were significantly reduced compared to E2_661_ vaccinated animals due to the contribution of HVR1 specificities [30]. The specificity of the antibody response was shown to be altered by the removal of all three variable domains with an increase in domain E and epitope III specificities observed in Δ123 vaccinated animals [30]. Key to the induction of bNAbs was the oligomeric status of Δ123 where sequentially higher molecular weight forms altered the antigenicity of the proteins through the sequential occlusion of the non-neutralizing face of E2 [30] and alteration of the exposure or folding of AR3 and CD81 binding site; domain D and E appeared unaltered in higher molecular weight forms of Δ123 compared to monomeric species. High molecular weight forms of Δ123 generated the highest titres of bNAbs towards G1-7, and significantly higher titres of antibodies towards domains E, D, AR3 and E2-CD81 blocking antibodies [30]. Furthermore, lower titres of non-NAbs towards domain A were elicited in animals vaccinated with high molecular weight forms of Δ123 compared to immune serum from animals vaccinated with monomeric forms of Δ123 [30]. 

To overcome the limitations of this approach related to the purification of high molecular weight forms from complex oligomeric mixtures expressed from transfected cells, a limited reduction and refolding approach was employed starting from a purely monomeric species of E2 [96]. Mutation of seven cysteine residues in E2 or Δ123 results in the expression of a purely monomeric form of E2 that retains CD81 binding and a majority of NAb domains [24]. Using either monomeric E2RBD or Δ123, or monomeric disulfide mutated E2RBD or Δ123 (E2RBDA7 and Δ123A7), Center et al. were able to assemble high molecular weight forms of Δ123 and Δ123A7; the presence of HVRs appeared to inhibit this assembly. Antigenically, the assembled Δ123 and Δ123A7 retained recognition by domain E and D antibodies and MAbs that recognized other linear epitopes on the neutralizing face of E2, but ablated recognition by non-NAbs directed to AR1 and domain A. In addition, the assembled Δ123 and Δ123A7 were highly stable even at 100 °C, resisting thermal denaturation unless a reducing agent was present [96]. 

A similar approach based on this discovery was pursued by He et al. [97] to generate a nanoparticle vaccine using E2core proteins lacking HVR1, a modified and truncated HVR2 (residues 460–484), and an extended igVR/VR3 deletion (residues 569–596) with an additional deletion in the beta-loop sandwich (residues 543–546) to reduce binding by non-NAbs with a truncated C-terminus (residue 645) in both G1a and G6a sequences [97]. The optimized constructs showed greater binding affinity for antibodies directed towards domain E, D, a subset of AR3 NAbs and AR2 however HepC3 and HepC74 binding affinity was similar or reduced, respectively. Deletion of residues 543–546 reduced binding by non-NAbs [97]. Structural examination of these constructs revealed that removal of HVR2 and VR3/igVR improves the stability of the E2 core while retaining the overall antigenic structure of the neutralizing face of E2 [97]. The optimized E2core proteins were engineered to form two different 60-mer nanoparticles of 34.5 and 37.5 nM diameter. The nanoparticle display of the optimized E2core enhanced recognition by all E2-specific antibodies tested with the exception of non-NAbs AR1B and E1 [97]. Vaccination of mice with 50–100 µg of G1a antigen four times in conjunction with AddaVax™ generated both autologous NAb and cross-neutralizing antibody responses to G2a and G5a viruses [97]. Assembly into a nanoparticle resulted in a 38-fold improvement in titres of antibody directed towards domain E compared to monomeric antigens, similar to what was observed by Vietheer et al. using HMW Δ123 antigens [97]. Nanoparticles displaying E2 have also been assembled using ferritin fused to E2 (residues 384–661) to form 10nM particles [98]. These nanoparticles had higher affinity for AR3 and domain E antibodies as well as CD81 compared to monomeric E2. Immunization resulted in the development of bNAbs to all 7 HCV genotypes tested with significantly higher levels to G1b, G2a, G4a and G5A viruses [98]. 

Both the intact E2 RBD and an E2 core domain lacking HVR1 and HVR2 have been assembled into lipid-based nanoparticles with mean diameters of 115 and 121 nM, respectively [99]. Antigenic analysis revealed that E2 core nanoparticles had enhanced binding to antibodies from AR1, AR2 and domain E, and similar levels of binding with AR3 specific antibody compared to intact E2RBD assembled into nanoparticles [99]. Immunization of mice three times with nanoparticle assembled E2 or monomeric E2 proteins with adjuvant revealed that nanoparticles were more immunogenic, generating higher antibody titres and NAbs [99]. Significantly, animals from the E2 core nanoparticle group displayed higher NAb activity to heterologous HCV-pseudotyped HIV particles than animals that received intact E2 nanoparticles, although their autologous NAb activity was lower, presumably because they did not generate HVR1 specific antibodies [99]. Together, the data support an approach wherein E2 is re-engineered to remove HVR1, HVR2 and VR3/igVR with additional mutations or modifications to suppress immunogenicity of non-NAb domains, either through assembly into high molecular weight forms, or mutagenesis to remove contact residues for non-NAbs. Immunogenicity studies conducted in guinea pigs with four vaccinations of 100 µg in AddaVax™, showed that the assembled proteins had similar immunogenic properties to native high molecular weight forms [96].

Glycoprotein E2 contains 11 N-linked glycans accounting for up to 50% of the molecular weight. The viral envelope proteins of pathogens such as HIV-1 are similarly highly glycosylated and glycans have been demonstrated to play a major role in both antibody recognition, and immune evasion through occluding access to epitopes on the underlying core domain and shifting of glycosylation sites on evolving mutants to circumvent immune recognition. The role of glycans on E2 in immune recognition has been examined in the context of both virus and recombinant soluble E2. Expression of E2_661_ in insect cells limits glycosylation to paucimannose type glycans, with less modification than that observed in mammalian cells [100]. Insect cell derived E2 was able to bind CD81, SR-B1 and was recognized by domain E and AR3 specific antibodies confirming it was correctly folded. In comparison to mammalian expressed E2, insect cell derived E2 demonstrated higher binding to AR3A, CD81 and SR-B1 and comparable levels of binding to domain E compared to mammalian cell expressed E2 suggesting that the extensive glycan modification observed in mammalian expressed E2 may occlude recognition by some ligands [100]. Vaccination with mammalian or insect cell derived sE2 revealed that insect cell derived E2 was superior with the induction of bNAbs to all 7 major genotypes, although the fine specificity of the immune serum was not further investigated [100]. In the context of insect cell expression, the immunogenicity of intact E2 and E2 lacking HVR1 was similar with induction of bNAbs in both cases and insect cell derived E2 protected against HCV challenge in an experimental mouse model of HCV infection [100].

Trivalent vaccines containing insect cell derived recombinant soluble E2 from genotypes 1a, 1b and 3a were used to immunize mice and macaques generating NAbs to all 7 HCV genotypes, and was superior to monovalent vaccine immunizations [101]. This approach shows that a multivalent vaccine could be a viable strategy for overcoming the genetic diversity of E2. Such an approach would also account for the possibility that epitopes might be present in different conformations based on genotype. Insect cell expression of a synthetic E2 consensus sequence for genotype 1 led to the expression of correctly folded CD81 binding competent E2, able to be recognized by conformation dependent antibodies. Immunization of guinea pigs generated potent homologous NAb to G1 viruses but failed to elicit bNAbs to G2a and G3a viruses [102]. 

### 3.2. Virus-Like Particle (VLP) Vaccines

Virus-like particle-based vaccines have been generated for a number of pathogens and some are in clinical use [103,104]. VLPs are highly attractive vaccine platforms as they can elicit both NAbs and cellular immune responses, they structurally resemble the wildtype virus from which they are derived, they contain no genetic material and so offer a safe and effective alternative to live, attenuated and inactivated vaccines [104,105]. Both Hepatitis B virus (HBV) and Human papillomavirus (HPV) vaccines have been developed from VLPs and are in clinical use in humans with ~95% and ~100% efficacy, respectively [106,107]. Various VLP platforms have been used to deliver HCV antigens including the p24/p27 surface protein of the HBV surface antigen (HBsAg-S) [108,109,110,111,112,113,114,115,116], a self-assembled core-E1-E2 of HCV itself [117,118,119,120,121,122,123,124,125,126], and genome devoid recombinant HCV-pseudotyped retrovirus derived VLPs [127]. Important considerations for the production of any vaccine are the ability to express the particle itself in a system that is amenable to future cGMP manufacture at high yields suitable for human clinical use, and retain the desired antigenicity of the heterologous components.

Chimeric HBV/HCV VLPs are constructed by including the E2 sequence into the self-assembling small envelope protein of HBV (HBsAg-S). One of the first studies constructed HBV/HCV chimeric VLPs displaying the genotype 1a and 1b HVR1 region of E2 within the ‘a’ determinant of HBsAg-S [112]. The particles were recognized by human antibodies from chronically infected people suggesting the HVR1 region was exposed on the external surface of VLPs. These particles were immunogenic in mice and induced genotype specific anti-HVR1 antibodies. Vaccination with a mixture of HVR1 G1a and HVR1 G2a HBV/HCV VLPs induced antibodies against both HVR1 epitopes and resulted in higher titres than those generated by immunization individually, suggesting a synergistic effect. Additionally, vaccination with the HVR1 G1a VLPs induced NAbs that were able to neutralize pseudotyped viruses containing both homologous and heterologous HVR1 regions [113].

Full length E1E2 has been efficiently incorporated into the HBsAg-S by replacing the N-terminal transmembrane domain of S with that of the HCV glycoprotein [115]. Particles were produced containing either E1, E2 or both E1 and E2, and immunization of rabbits generated a strong antibody response. The neutralization potential of this serum was assessed against G1a, G1b, G2a and G3a viruses and serum from animals vaccinated with VLPs containing E2, or E1 and E2 showed the strongest virus neutralization against G1a and G1b, and weaker neutralization observed towards G2a and G3a [108]. Subsequently, it was found that immunization with HBsAg-S VLPs containing E1 alone or E2 alone was superior both in terms of titre and neutralization activity than immunization with HBsAg-S VLPs containing both E1 and E2 [109]. Combining the serum from animals vaccinated with HBsAg-S VLPs containing E1 or E2 synergistically enhanced neutralization relative to the neutralization observed with E1 or E2 antiserum alone [109].

The HCV structural proteins core, E1 and E2 are able to self-assemble into HCV-like particles (HCV-LP) within insect cells but remain intracellular. Extraction of the HCV-LPs revealed that they were recognized by E1 and E2 specific MAbs and serum from infected people, suggesting some antigenic features were retained and as the particles were produced in insect cells, glycosylation of both E1 and E2 was high mannose [118]. Immunization of mice showed an anti-core and anti-E2 specific humoral response and proliferative T cell response towards core, E1 and E2 [118,119,121]. The immunogenicity of the HCV-LPs was reduced when denatured suggesting that the conformation of proteins and presentation in the form of a VLP were critical to immunogenicity [119]. Immunization of chimpanzees with 4 doses of HCV-LPs resulted in strong CD4+ and CD8+ responses to core, E1 and E2, and challenge with homologous G1b virus resulted in transient viraemia with lower peak viral loads and all four animals were HCV negative one year following challenge compared to placebo vaccinated animals of which ¾ developed chronic infection [117]. Interestingly, none of the vaccinated chimpanzees developed anti-HCV antibodies during the vaccination period suggesting that the major contributor to blunting HCV viraemia in this case was T cell driven. At the time of conducting these studies, extensive tools for investigating the contribution of B cell immunity were not available and future studies could revisit the use of HCV-LPs and whether B cell immunity is stimulated and contributes to vaccine induced protection. 

An alternative platform for the production of HCV-LP employs an adenovirus encoding the structural genes of HCV. In the initial studies the particles were designed to incorporate G1a H77c core, E1 and E2 [123,124]. Production of the VLPs in Huh-7 cells generated 40–80 nM VLPs that contained ApoE and ApoC, high mannose E1 and E2 glycoproteins and core, and were able to interact with CD81. Mice immunized with adenoviral derived HCV-LPs generated a strong E2 specific antibody response with associated neutralization activity and were able to stimulate the production of HLA A2 specific T cell responses to HCV core in MHC class 1 transgenic mice. In order to broaden the immune response to provide protection against multiple HCV genotypes, subsequent studies expanded the adenovirus derived VLPs to include G1b, G2a and G3a VLPs [125,126,128]. These VLPs were recognized by human MAbs specific to domain B and D but not non-NAbs towards domain A. Particles were only weakly recognized by antibodies specific to domains E and C. These data suggest that while some domains of E2 are correctly folded and/or accessible for binding by neutralizing antibodies, other important neutralization domains may be occluded and could restrict the generation of NAbs [128]. Pigs vaccinated with quadrivalent vaccine produced cross-neutralizing antibodies towards G1a, G2a and G3a [129]. In mice, antigen specific B cells, IFN-gamma secreting CD4+ and CD8+ T cells, and granzyme B producing T cells were elicited [126]. Overall, the platform provides a pathway to the production of multivalent vaccines that generate both neutralizing antibody and cellular immune responses. 

### 3.3. Inactivated HCV Particle Vaccines

Inactivated vaccines are produced by growing large quantities of virus in cell lines, followed by an inactivation process to disable virus replication. A key benefit of such systems is that the vaccine contains all the structural components of the virion and these structural components can maintain aspects of their native conformation. In the case of HCV, the native E1E2 glycoprotein complex would be presented to the immune system and theoretically allow the generation of AR4-like and AR5-like specificities in addition to specificities resident on the RBD alone, if they are exposed. A vaccine candidate using inactive HCVcc particles was developed by producing chimeric G2a J6/JFH-1 cell culture generated HCV [130]. Immunogenicity studies in mice revealed the generation of anti-E1 and anti-E2 antibodies, with the ability to neutralize G1a, G1b and G2a.

Passive transfer of the serum vaccinated animals could protect against low dose (10 × 3 RNA copies) HCV challenge in human liver chimeric uPA-SCID mice, however, animals challenged with higher viral doses (10 × 5 RNA copies) were not protected. A subsequent study examined immunogenicity of inactivated HCV vaccine in marmosets, generating potent humoral and cellular immune responses [131]. The animals vaccinated with HCV particles combined with a novel nanoparticle adjuvant, induced antibodies reactive to E2 and core and with the ability to neutralize G1a, G1b, G2a and G3a viruses. Cellular responses were also detected by measuring IFN-gamma mRNA transcripts in PBMCs and splenocytes. While this technology has distinct advantages as it presents a near-native like inactivate particle to the immune system allowing the generation of both antibody and cell-based responses to all structural components of the vaccine, manufacturing such a vaccine at large scale remains challenging given the relatively low yields of HCV in cell culture. In depth antigenic assessment of such vaccines has not been conducted, nor has the immune response been analyzed in detail to determine the fine specificity of antibodies.

### 3.4. Viral Vector Vaccines

Viral vectors are effective tools to deliver the genetic sequence for foreign antigens into target cells and can induce both cellular and humoral immunity against infectious diseases. Viral vector platforms include adenoviruses, vesicular stomatitis virus, modified Vaccinia Ankara (MVA), influenza, alphavirus and paramyxovirus vectors [132]. Furthermore, novel adenoviral vectors with low seroprevalence in the human population include the chimpanzee adenoviruses (ChAd). The major advantages of viral vector systems are that they allow for the insertion of a genetic sequence into the vector which can be engineered to contain desired epitopes, and modifications within those epitopes, with immunization resulting in the endogenous expression of the encoded proteins, generating potent CD4+ and CD8+ responses. In addition, inclusion of a B cell immunogen can result in the generation of antibodies. Derived from the vaccinia virus, MVA vaccines are growth adapted to avian cells and have lost the ability to replicate in mammalian hosts. It is an excellent vaccine vector due to its safety and ability to boost immune responses by broadening and increasing the magnitude of pre-existing T cell responses. One consideration for using viral vectored approaches is the generation of antibodies towards the viral vector proteins themselves which can suppress the ability to boost immune responses after the 1st vaccination. To overcome this limitation, heterologous prime-boost strategies can be used employing different viral vectors or delivery systems for prime and boost strategies. 

Proof of concept that viral vectors can generate potent T cell responses was first demonstrated in chimpanzees using rare serotypes of adenovirus (Ad) Ad6 and Ad24 encoding the entire HCV non-structural region (NS3 to NS5B of G1b, BK strain) [133]. Animals received a heterologous Ad6/Ad24 prime boost, followed by a DNA prime encoding the same HCV sequence. Functional T cells were generated in all animals with potent, broad and cross-reactive cytotoxic T cell responses generated in 4/5 animals. The chimpanzees were challenged with a heterologous G1a strain where four animals showed a brief duration of acute viremia followed by viral clearance, associated with an expansion of CD8+ IFN-gamma T cells. Following the results from this study, a phase I trial in humans was conducted in which participants were primed with Ad6 or chimpanzee adenovirus 3 (ChAd3) viral vectors expressing the non-structural genes 3–5B of HCV (Ad6-NSmut or ChAd3-NSmut), followed by a boost with either Ad6-NSmut or ChAd3-NSmut [134]. The vaccine was shown to be safe and in patients who received the highest dose it induced a broad and multi-specific CD4+ and CD8+ cellular immune response. In another phase I human trial, a heterologous prime/boost vaccination strategy was investigated [135]. Healthy volunteers were immunized with ChAd3-NSmut and boosted with MVA carrying the same HCV sequence (MVA-NSmut) The results of this trial showed that HCV-specific T cells induced by ChAd3NSmut are optimally boosted with MVA-NSmut and generate very high levels of both CD8+ and CD4+ HCV-specific T cells targeting multiple HCV antigens. Sustained memory and effector T cell populations were generated, and T cell memory evolved over time and was improved by the heterologous MVA-NSmut boost. This ChAd/MVA vaccine regimen progressed to a Phase II clinical trial (NCT01436357) enrolling 274 HCV-naïve injecting drug users at high risk for infection. Whilst the vaccine was safe, and T cell responses were elicited in 78% of vaccinated participants, the vaccine failed to show efficacy in preventing chronic HCV, although peak viral loads were lower in the vaccine group [136]. The lack of efficacy could be explained by a mismatch between the vaccine antigens and circulating HCV in the enrolled injecting drug user community, or due to insufficient immunity. Further optimization of the antigens and inclusion of antigens designed to generate humoral immunity may be advantageous in future studies.

To address the issue of antigen diversity amongst circulating strains, Von delft et al. applied a computer algorithm to identify genomic regions that are conserved between HCV subtypes and genotypes. Three sequences were designed to encode conserved regions between G1a and G1b, G1 and G3 and G1–6, including epitopes associated with spontaneous resolution of HCV [137]. The chimeric HCV sequence was encoded in a serotype 1 ChAd vector (ChAdOx1) and the immunogenicity evaluated in mice. Strong HCV specific CD4+ and CD8+ T cell responses were generated, cross reactive with G1a, G1b and G3a peptides. It is not yet known if this second-generation T cell immunogen similarly broadens HCV specific T cell cross-reactivity in outbred animals or humans. However, ChAdOx1 has been successfully used as a vaccine for COVID-19 to deliver the spike antigen of SARS-COV-2 and generates high titre NAb and strong T cell responses with vaccine efficacy in humans against matched virus isolates ranges from 62–90% after two doses [138,139]. This suggests that viral vectors can be successfully used to deliver both B and T cell immunogens and could be employed for other pathogens such as HCV. 

Incorporation of the structural region of HCV into adenovirus vectors has been explored using adenovirus serotype 6 in various prime-boost strategies either vector or protein based [140]. Adenovirus 6 based vectors either encoded G1b truncated E2 (residues 384–662) or the entire G1b E1E2p7 region. Both mice and guinea pigs generated an antibody response after immunization and developed homologous NAb with some evidence of cross-neutralization against diverse HCV genotypes [140]. 

Other viral vectors such as influenza virus have potential in the HCV vaccine development. Zhang et al. rescued a recombinant influenza virus vector which encoded HCV core, E1 and E2 epitopes in the PR8 NS1 (rgFLU-HCV_CE1E2_) gene [141]. Intranasal inoculation of mice with rescued viral particles induced neutralizing antibodies towards homologous HCV virus and Th1 and Th2 cell response to HCV antigens. Overall viral vectors are promising delivery methods for both T and B cell epitopes of HCV. 

### 3.5. Synthetic Peptide Vaccines

Peptides corresponding to predicted immunogenic regions of a pathogen can be used individually or as a pool or array of multiple peptides targeting multiple epitopes. Advantages of peptides for vaccination is that the immune response can focus on a desired epitope, peptides do not contain infectious material and there is no risk of reversion or integration of genomic material into the recipient, peptide vaccines are easily modified to improve stability and large-scale production is highly economic. However, there are currently no peptide vaccines licensed for use in humans. One of the main challenges encountered with peptide vaccines is poor immunogenicity and the need for potent adjuvants to stimulate T helper cells, which are required for both efficient generation of T cell and B cell immunity. Peptide vaccines are further limited by HLA-restriction limiting which epitopes are presented to the immune system. In addition, as peptide sequences are usually relatively short (<30 amino acids), conformational epitopes and complex epitopes requiring extensive tertiary or quaternary structure are not well presented. Advanced techniques such as cyclisation can be used to improve the conformational presentation of B cell epitopes. Many peptide vaccine approaches for HCV have focused on the induction of T cell immune responses using short peptides. However, approaches targeting B cell responses have begun to emerge using peptides corresponding to non-conformational epitopes or making use of cyclization or scaffolding techniques [142,143,144,145,146]. 

A HCV peptide vaccine based on a cyclic defensin protein has been employed to mimic the beta-hairpin conformation of domain E of the glycoprotein E2 of HCV [143]. Vaccination of mice with the cyclic peptide was superior to a linear peptide of the same region. It induced domain E specific antibodies and neutralized homologous G1a HCVpp but only limited cross-neutralization was observed [143]. In addition, a bivalent E2 core domain was engineered that displayed two copies of domain E per molecule. The bivalent antigen was superior to the cyclic peptide in induction of NAb and their breadth of neutralization, likely due to the presence of additional domain E epitopes [143]. Other examples of peptide vaccines for HCV include, a multi-epitope peptide vaccine, encoding conserved regions between G2a and G4a including E1 amino acids 315–326, domain E, the CD81 binding loop and T cell epitopes from NS4B, NS5A and NS5B. The peptides were synthesized in the form of a multiple antigenic peptide and administered to BALB/c mice at different concentrations to investigate humoral and cellular immune responses [145]. The vaccine was found to generate antibodies specific to each peptide, NAbs towards G2a and G4a viruses, and a persistent cellular response in mice [145]. 

Grollo et al. used patient serum/plasma obtained from an HCV infected individual to identify antigenic peptides using an overlapping 18-mer peptide library spanning the E1E2 region [147]. Twenty-one peptides were identified, 7 within E1 and 14 in E2, 4 of which were within HVR1 and 5 spanning residues 602–648. Four of the peptides were conserved across genotypes and selected for immunogenicity studies. One of these peptides was within E1, while 3 were within E2. An additional peptide within the N-terminus of HVR1 was also selected for analysis. Antigenic peptides were then used as immunogens to generate antibodies in mice, and 4/5 peptides generated homologous NAbs. Interestingly, none of the 4 non-HVR1 peptides were located with the RBD, transmembrane domain or conserved heptad repeat region, and represented novel antigenic sites not usually associated with NAbs. The approach could be adapted to search for novel specificities using immune serum from individuals who spontaneously cleared HCV infection to identify domains associated with protection against chronic infection. 

Bivalent HCV peptide vaccines have been shown to elicit pan-genotypic NAbs in mice [146]. A novel method for sequence optimization utilized an in silico approach to determine two optimal sequence clones of the HVR1 region of glycoprotein E2. The two peptides were administered as a single peptide each and in a bivalent vaccine to three groups of Balb/c mice. All three approaches elicited high-titre IgG and the bivalent vaccine was found to elicit antibodies exclusively targeting the more conserved N-terminus of the HVR1 region [146]. Upon analysis of the neutralization capacity of sera from the three groups despite the vaccine being based of a mono-genotypic sequence, broad, cross-genotypic neutralization could be observed [146]. The ability to narrowly focus immune responses with peptide vaccines suggest they might best be used in prime boost approaches with other vaccine strategies to enhance the immune response to otherwise subdominant epitopes. 

### 3.6. DNA and RNA Vaccines

An alternative approach to protein and viral vectored vaccine modalities is the delivery of genetic information of the antigen in the form of either DNA or messenger RNA (mRNA). Potential advantages of this approach are the endogenous expression of the viral protein allowing the generation of CD4+, CD8+ and humoral immune responses, the antigens are synthesized in situ using host cell machinery including host-cell derived post-translational modifications giving rise to a “native-like” presentation to the immune system, and large-scale cGMP manufacturing is scalable and cheap. The first DNA vaccines tested in human trials was an HIV-1 vaccine in 1998 encoding the *env* and *rev* genes [148]. Since then, DNA vaccines have been widely explored against many pathogens including malaria, HBV and influenza with some entering clinical trials [149,150,151]. While currently there are no DNA vaccines approved for use in humans, some DNA vaccines have been approved for veterinary use against west nile virus in horses [152] and canine melanoma [153].

The advent of COVID-19 has seen mRNA vaccines rapidly advance through human clinical trials, and the first mRNA vaccines licensed for use in humans from Moderna and Pfizer/BioNTech. mRNA is an intermediary between DNA and protein translation and has been investigated as a delivery strategy since 1990 [154]. Hampering the use of mRNA was delivery of the mRNA into the cell, instability of the RNA and poor translation resulting in low level gene expression and over stimulation of the innate immune response in response to foreign RNA. These limitations were overcome with the use of synthetic nucleosides, optimization of regulatory elements, codon optimization, improvements in purification methods and lipid nanoparticle delivery systems and has been extensively reviewed [155].

As both E1 and E2 normally localize to the endoplasmic reticulum, inclusion of their transmembrane domain regions in a DNA or mRNA vaccines is not expected to efficiently stimulate humoral immunity. One of the strategies used to increase the immunogenicity of E2 has been to either delete the TMD or modify it to favor secretion or cell-surface expression, respectively. A DNA vaccine encoding truncated E2 (residues 384–715) chimerized with the transmembrane region of platelet-derived growth factor receptor (PDGFR) to facilitate surface expression was used to immunize mice and macaques [156]. All animals developed a strong humoral immune response which was markedly higher that what was observed using DNA encoding E2 retained intracellularly. This candidate was then used to immunize two chimpanzees who received three immunizations followed by experimental challenge with homologous HCV three weeks later. One chimpanzee developed anti-E2 HVR1-specific antibodies, but no cellular immunity was detected prior to challenge, while the second chimpanzee had low levels of anti-E2 antibodies but had a measurable CD4+ T cell response prior to challenge. Both chimpanzees became infected but cleared virus faster than the non-vaccinated control [157]. 

Rollier et al. also aimed to induce both humoral and cellular responses with a multicomponent vaccine strategy consisting of DNA plasmids that expressed E1, E2, core and NS3 [158]. Two chimpanzees were immunized three times with G1a DNA constructs and a further four times with recombinant E1, E2, core and NS3 proteins of G1b, followed by challenge with heterologous G1b HCV. Following DNA immunization, a strong anti-E2 antibody response was generated, but antibodies towards E1, core and NS3 were not observed until immunization with protein. The vaccine was able to accelerate viral clearance in one animal within four weeks, while the second animal demonstrated control of viremia [158]. The regimen was then further modified to replace protein vaccinations with MVA vectors encoding core, E1, E2 and NS3, followed by challenge with G1b [159]. Detectable NAb titers were not induced and only one animal cleared infection [159]. The authors then designed a vaccine regimen in which they employed four vectors: DNA, MVA, Semliki forest virus (SFV) and human adenovirus (Ad5) to express the three structural proteins and NS3 and evaluated immunogenicity in rhesus macaques in various prime-boost combinations. Vaccination with DNA prime/Ad5 boost induced superior E2 specific antibody responses but none of the combinations induced NAbs or generated core, E1, E2 and NS3 specific T cell responses [160].

## 4. Conclusions

Developing effective vaccination strategies for HCV presents a unique challenge, due to its antigenic heterogeneity and immune evasion mechanisms. Other antigenically diverse chronic viruses such as HIV-1 have explored the use of mosaic vaccines [161,162], polyvalent antigens [163,164], or heterologous prime boost strategies [165] to broaden both T and B cell immunity and serve as useful examples on which to base future strategies for HCV vaccine development. Considerable knowledge has now been gained to define key sites of vulnerability for antibody mediated neutralization on the E1E2 glycoprotein complex. Whilst these vaccines can be tested in animal challenge models of HCV infection, none of these models exactly recapitulate the complexity of HCV infection in humans, nor the human immune response to vaccines. Further advances in HCV vaccine development are likely to only come by testing vaccines in experimental phase I human studies to determine their safety and immunogenicity. Recently, there has been consideration for a human challenge model of HCV infection [166]. This would allow rapid assessment of multiple vaccine candidates at pilot scale, reducing costs and de-risking further vaccine development in larger phase II and III studies. COVID-19 has substantially changed the vaccine landscape and may foreshadow a new era in HCV vaccine development.

## Figures and Tables

**Figure 1 viruses-13-00805-f001:**
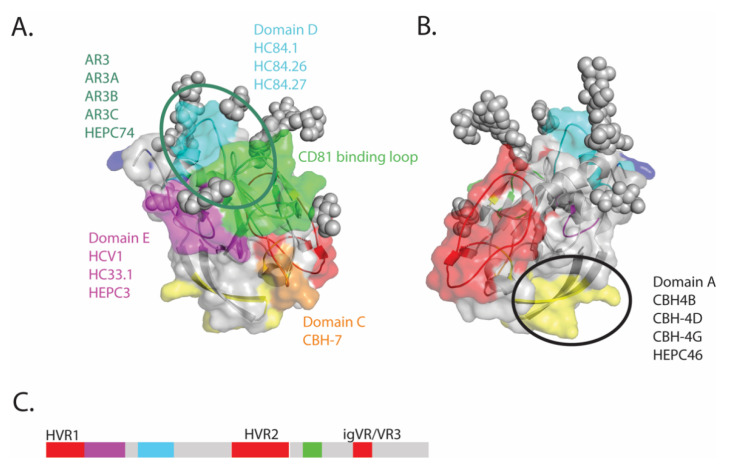
The antigenic organization of HCV glycoprotein E2 (PDB: 6MEJ) [25]. The front face of E2 (**A**) shows the location of domain E/epitope I (magenta), domain D/epitope II (cyan), domain B/AR3 (cyan/green), and domain C (orange). HVR2 and igVR/VR3 are shown in red. A 180° rotation of E2 (**B**) shows the location of HVR2 and the igVR/VR3 in red and regions contributing to non-NAb epitopes AR1 and domain A are shown in yellow. Glycans are depicted as spheres. (**C**) Schematic representation of the organization of the E2 ectodomain with locations of HVR1, HVR2 and igVR/VR3 (red), linear antigenic domains E (magenta) and D (cyan) and the CD81 binding loop (green).

## Data Availability

No new data were created or analysed in this study. Data sharing is not applicable to this article.

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
