# Peer review of "To Include or Occlude: Rational Engineering of HCV Vaccines for Humoral Immunity"

_viruses, 2021, doi:10.3390/v13050805_

Round 1
Reviewer 1 Report
This is a comprehensive review and provide useful insights for the development of HCV vaccines to target E1E2 based on the antigenic structure and relevant preclinical and clinical data. In addition to the critique that has been provided on the published studies, the review could benefit if it engaged in a more thorough discussion about the feasibility and possibility of developing effective vaccination strategies that can elicit protective humoral and T cell immunity given the antigenic diversity. Also, there could be more information about polyfunctional antibodies (e.g. ADCC and ADCP) and their role in controlling HCV infection especially given that Ad26/MVA/Env mosaic vaccination strategy for HIV elicited such responses and is being tested in Phase III for HIV. In general, clues from vaccine development for antigenically diverse, chronic virus infections could be incorporated especially since no vaccine has entered Phase III for HCV. Minor point: please check throughout that in text referencing is performed consistently. For example, in text referencing is missing for several sentences in the paragraph starting from line 532.
Author Response
Thank you for your time in reviewing our manuscript and providing valuable comments. We have addressed each point as follows:
- more thorough discussion about the feasibility and possibility of developing effective vaccination strategies that can elicit protective humoral and T cell immunity given the antigenic diversity.
We thank the reviewer for this insightful comment. We have modified the text at Lines 960 to discuss this perspective:
“Developing effective vaccination strategies for HCV presents a unique challenge, due to its antigenic heterogeneity and immune evasion mechanisms. Other antigenically diverse chronic viruses such as HIV-1 have explored the use of mosaic vaccines (1, 2), polyvalent antigens (3, 4), or heterologous prime boost strategies (5) to broaden both T and B cell immunity and serve as useful examples on which to base future strategies for HCV vaccine development.”
- In general, clues from vaccine development for antigenically diverse, chronic virus infections could be incorporated especially since no vaccine has entered Phase III for HCV.
We have addressed this comment briefly in point one above. The purpose of this review was to focus on how to modify E1 and E2 antigens to improve the generation of bNAbs and review the current literature in this area. Whilst it would be interesting to explore what has been performed for the surface proteins of Influenza and HIV, we feel this is out of scope for this review.
- there could be more information about polyfunctional antibodies (e.g. ADCC and ADCP) and their role in controlling HCV infection especially given that Ad26/MVA/Env mosaic vaccination strategy for HIV elicited such responses and is being tested in Phase III for HIV
Thank you for this suggestion. There is limited knowledge of the role of ADCC for HCV, partly because E1 and E2 are retained in the endoplasmic reticulum and so it is unclear whether cell-mediated clearance can occur. We have added the following at line 416:
“2.4 Polyfunctional Antibodies in HCV infection
In addition to their neutralizing capabilities, antibodies can also mediate other functions through their Fc domain. These include antibody – dependent cellular cytotoxicity (ADCC). Very limited studies are available on the role of such polyfunctional antibodies and ADCC in HCV infection. Studies of HCV-infected individuals have shown the presence of anti-E2 antibodies capable of mediating ADCC (6). However, these types of antibodies were more prevalent in chronic infection than in acute or self-limited infection, arguing against a contribution to resolution of infection (6). Given that the hepatitis C E1 and E2 glycoproteins are strongly retained in the endoplasmic reticulum during biosynthesis, it is not whether ADCC plays a major role in immune clearance of HCV infection. Further studies are warranted to resolve this question.”
- Referencing is missing for several sentences in the paragraph starting from line 532.
We have updated the referencing from line 532.
Reviewer 2 Report
Schlotthauer et al. reviewed the extensive data about the humoral response to HCV glycoproteins E1 and E2. This knowledge is important for designing the prophylactic vaccine against HCV. I learned a lot and the material discussed is interesting. It would be of interest to many readers both within or outside the field of studying HCV.
I have few minor comments on the manuscript:
- HCV is divided into 7 major genotypes (6 common ones, plus 1 uncommon one but distinct genotype 7a). Authors referred to HCV exist as 8 genotypes (line 32 and line 37), it would be of interest to include the reference for it.
- In line 106, It is stated that the number referred is according to prototype H77 numbering. The amino acid number has already mentioned in the earlier section (line 85). Perhaps the numbering notations in line 106 should be moved to line 85.
- In line 126, Terminology on the antigenic domains (e.g. Domain E) is used to describe the atomic structure of E2. Perhaps more annotations are needed for people outside the field of HCV, such as refer to Figure 1.
- In section 2.3, there is a section on eliciting non-Nab. This might be a section to include work(s) (e.g. Kachko et al. Hepatology 2015) related to interfering antibodies that could reduce the efficacy of neutralizing antibodies.
Author Response
Thank you for your time in reviewing our manuscript, we appreciate your comments and have addressed each individually as follows:
- HCV is divided into 7 major genotypes (6 common ones, plus 1 uncommon one but distinct genotype 7a). Authors referred to HCV exist as 8 genotypes (line 32 and line 37), it would be of interest to include the reference for it.
We have added in the reference - Borgia, et al., (2018) to the manuscript. This study identifies HCV genotype 8 and confirms its circulation in the population.
- In line 106, It is stated that the number referred is according to prototype H77 numbering. The amino acid number has already mentioned in the earlier section (line 85). Perhaps the numbering notations in line 106 should be moved to line 85,
Thank you. This has now been modified and have moved this statement to line 86 as suggested.
- In line 126, Terminology on the antigenic domains (e.g. Domain E) is used to describe the atomic structure of E2. Perhaps more annotations are needed for people outside the field of HCV, such as refer to Figure 1.
Thank you for this comment. We have added the following sentence at line 128: “The antigenic domains of E2 are shown in figure 1 and discussed in detail in section 2.2.”
- In section 2.3, there is a section on eliciting non-Nab. This might be a section to include work(s) (e.g. Kachko et al. Hepatology 2015) related to interfering antibodies that could reduce the efficacy of neutralizing antibodies
Thank you for this suggestion. We have added a section at line 406 and provided examples;
“Additionally some antibody specificities can interfere with the neutralizing activity of other NAbs. A study found that immunization of healthy adults with recombinant E1E2 can induce both epitope I and epitope II specific antibodies. Removal of epitope II specific NAbs from the immune serum increased the neutralization activity by up to 4.9 fold (7) . This was also observed in chimpanzees immunized with E1E2, whereby removal of epitope II specific antibody not only enhanced neutralization activity of serum, but also revealed a cross neutralizing antibody response (8). This suggests that while both epitope I and epitope II specific antibodies are induced, not all epitope II directed antibodies are indeed neutralizing or beneficial to neutralization activity of serum. “